# Hemodynamic Melody of Postnatal Cardiac and Pulmonary Development in Children with Congenital Heart Diseases

**DOI:** 10.3390/biology13040234

**Published:** 2024-03-31

**Authors:** Sixie Zheng, Lincai Ye

**Affiliations:** 1Department of Thoracic and Cardiovascular Surgery, Shanghai Children’s Medical Center, School of Medicine, Shanghai Jiao Tong University, National Children’s Medical Center, Shanghai 200127, China; zhengsixie15@sjtu.edu.cn; 2Shanghai Institute for Pediatric Congenital Heart Disease, Shanghai Children’s Medical Center, School of Medicine, Shanghai Jiao Tong University, National Children’s Medical Center, Shanghai 200127, China

**Keywords:** volume overload, pressure overload, cardiomyocyte, maturation, proliferation, pulmonary alveolar dysplasia, pulmonary blood flow

## Abstract

**Simple Summary:**

Abnormal hemodynamics and cardiac and pulmonary development intertwine to form the most important features of children with congenital heart diseases (CHDs). Here, we review the varieties of hemodynamic abnormalities that exist in children with CHDs and the recently developed neonatal rodent models of CHDs, which have inspired and brought us new ideas and inspirations in cardiomyocyte proliferation and maturation, as well as in alveolar development.

**Abstract:**

Hemodynamics is the eternal theme of the circulatory system. Abnormal hemodynamics and cardiac and pulmonary development intertwine to form the most important features of children with congenital heart diseases (CHDs), thus determining these children’s long-term quality of life. Here, we review the varieties of hemodynamic abnormalities that exist in children with CHDs, the recently developed neonatal rodent models of CHDs, and the inspirations these models have brought us in the areas of cardiomyocyte proliferation and maturation, as well as in alveolar development. Furthermore, current limitations, future directions, and clinical decision making based on these inspirations are highlighted. Understanding how CHD-associated hemodynamic scenarios shape postnatal heart and lung development may provide a novel path to improving the long-term quality of life of children with CHDs, transplantation of stem cell-derived cardiomyocytes, and cardiac regeneration.

## 1. Introduction

Since Harvey’s discovery of the circulatory system in 1628, it has been recognized that the primary function of the circulatory system is to provide organs, including the heart and lungs, with well-oxygenated blood and nutrients, as well as to transport all types of harmful metabolites [1,2]. The hemodynamics resulting from blood flowing in the circulatory system inevitably becomes the eternal theme of the circulatory system [1,2]. Insufficient embryonic volume load is one of the main causes of left ventricular dysplasia, while adult volume overload (VO) or pressure overload (PO) lead to cardiac failure [3,4,5,6,7]. However, pediatric hearts and lungs are in a stage of active development. For example, 90% of alveoli form between the ages of 0 and 7 years, and there is an extensive cardiomyocyte maturation transition [8,9]. Hemodynamic modifications of embryonic or adult hearts and lungs are well established [3,4,5,6,7] but leave a vacuum to be filled in our understanding of the hemodynamic shaping of pediatric hearts and lungs.

Congenital heart diseases (CHDs) represent the most prevalent birth defect in the world [10,11]. With the progress in pediatric cardiac surgery, cardiology, and cardiac intensive care, most simple CHDs can be corrected, and the overall mortality shows a significant downward trend. However, the mortalities of complex CHDs, such as hypoplastic left heart syndrome, Ebstein anomaly (EA), and tetralogy of Fallot (TOF), are still very high. Most of these cannot be physiologically corrected, leaving residual hemodynamic abnormalities, such as cardiac volume overload (VO), pressure overload (PO), pulmonary congestion and reduced pulmonary blood flow (RPF); therefore, hemodynamic abnormalities dominate postnatal cardiac and pulmonary development in children with CHDs [12,13,14,15,16,17]. As a result, the long-term life quality of children with complex CHDs is poor, with significantly reduced exercise capacity, impaired cardiac performance, and some children ultimately requiring heart or lung transplantation [18,19].

In addition, cardiovascular therapies that are very effective in adults have limited effectiveness or even cause harm when treating infants and young children [20,21]. A possible reason underlying this phenomenon is that the extremely small size of a neonatal mouse heart challenges the creation of neonatal surgical mouse models of cardiovascular diseases. Consequently, clinicians or scientists generally use adult mouse models to explore mechanisms and pathophysiology and to obtain targets for pediatric cardiovascular diseases.

Here, we have reviewed the abnormal hemodynamics in children with CHDs and the recently developed neonatal surgeries for producing CHD-hemodynamic-associated animal models, which include neonatal aortic and inferior vena cava fistula (nACF) surgery, neonatal pulmonary artery banding (nPAB) surgery, neonatal pulmonary vein banding (nPVB) surgery, and neonatal transverse aortic constriction (nTAC) surgery [22,23,24,25,26,27,28,29,30,31,32,33]. nACF produces atrial and ventricular VO and pulmonary congestion; nPAB produces right ventricular (RV) PO and RPF; nPVB produces pulmonary vein stenosis (PVS), which is one of the most challenging issues of pediatric CHDs [34]; and nTAC produces left ventricular (LV) PO. These neonatal surgeries greatly inspire our understanding of the hemodynamic melody of postnatal cardiac and pulmonary development, such as cardiomyocyte proliferation and maturation and also pulmonary dysplasia.

## 2. Abnormal Hemodynamics in Children with CHD

### 2.1. Abnormal Hemodynamics in Ventricles

CHD usually leads to ventricular VO and PO, specifically RV VO, LV VO, RV PO, and LV PO. Table 1 summarizes each type of abnormal hemodynamics and its corresponding CHDs.

There are three types of CHDs that can result in RV VO. The first type is CHD with a left-to-right shunt, which includes atrial septal defect (ASD), patent ductus arteriosus (PDA), partial anomalous pulmonary vein drainage (PAPVD), atrioventricular septal defect (AVSD), and ventricular septal defect (VSD) [17,35,36,37,38]. The second type is right-heart-valve regurgitation-associated CHDs, which include tricuspid regurgitation (TR), EA, pulmonary regurgitation, pulmonary valve insufficiency (PVI), and TOF after patch enlargement of the RV outflow tract correction [39,40,41,42,43]. The third includes advanced pulmonary hypertension (PH), increased returning blood (e.g., in thyrotoxicosis), and functional univentricular conditions (e.g., hypoplastic left heart syndrome and Fontan surgery) [44,45,46].

Right-to-left shunting and left-heart-valvular regurgitation can lead to LV VO. Right-to-left shunting-associated CHDs include ASD with tricuspid stenosis (TS) or RV outflow obstruction (RVOT) [47]. These defects result in a high pressure in the right atrium compared to the left atrium, leading to a right-to-left shunt. Left-heart-valvular regurgitation includes congenital mitral regurgitation (CMR), acquired mitral regurgitation (rheumatic or calcific mitral valve), aortic valve calcification, and bicuspid aortic valve (BAV) [48,49,50]. In addition, increased venous return can lead to LV VO.

PH, left cardiac defects that cause PH at later stages, and RVOT obstructive diseases may cause RV PO. CHDs that cause PH include PVS, large VSD, and PDA [34,35,38,51]. Large VSD and PDA lead to increased pulmonary blood flow, resulting in pulmonary congestion and small vessel remodeling, which in turn cause PH. CHD with a left cardiac defect that causes PH includes mitral or aortic stenosis (MS or AS) and coarctation/interruption of the aortic arch (CAA/IAA) [49,50,52]. CHDs with RVOT obstruction include TOF, isolated pulmonary artery obstruction or embolism, pulmonary artery stenosis (PAS), and pulmonary valve stenosis (PS) [53,54,55,56].

Hypertension and left ventricular outflow tract (LVOT) obstruction lead to LV PO. Hypertension is uncommon in children. CHD with LVOT obstruction includes BAV, CAA/IAA, mitral or aortic stenosis, and hypertrophic cardiomyopathy [49,50,52,57].

It is evident from Table 1 that various hemodynamic abnormalities, including both ventricular VO and PO, usually coexist in the same type of CHD. In addition, these hemodynamic abnormalities are closely associated with disease progression, and sometimes one type of abnormal hemodynamic abnormality dominates a particular stage of disease progression. The extents of VO and PO often determine the performance of the RV and LV, as well as surgical approaches and the prognosis of children with CHDs. Therefore, gaining a comprehensive understanding of how VO and PO shape postnatal cardiac development may help establish a theoretical foundation for enhancing the quality of life in children with CHD.

### 2.2. Abnormal Hemodynamics in Lungs

A left-to-right shunt or increased venous return will induce increased pulmonary blood flow (IPF), which leads to pulmonary congestion, ultimately resulting in pulmonary hypertension and congestive heart failure. RVOT obstructions, such as TOF and PS, result in RPF, which leads to pulmonary dysplasia, reduced exercise endurance, and renders patients less susceptible to COVID-19 infection [25,29,58,59]. A deeper understanding of the mechanisms by which IPF and RPF regulate postnatal lung development may help improve lung function and exercise capacity in children with CHD.

PVS is another deadly CHD [28,34]. Stenosis of the pulmonary vein leads to the remodeling of small pulmonary vessels, which in turn leads to PH, and ultimately right heart failure occurs. Despite advancements in surgical techniques, interventional procedures, and postoperative monitoring, the prognosis for children with PVS remains unfavorable, with 60% of children with PVS dying within 2 years after diagnosis [34]. Moreover, due to stenosis, there may be oozing blood upstream of the pulmonary vessels, giving rise to hemoptysis and pulmonary infections, which further deteriorate the prognosis of children with PVS. Understanding how and why PVS occurs is crucial to treating children with PVS.

## 3. Immature Hearts and Lungs in Children

### 3.1. Immature Hearts in Children

Unlike adult mature hearts and lungs, children’s hearts and lungs are immature and at an active stage of development [8,12].

To meet the physiological demands of pumping blood at adulthood, there are three main characteristics that immature cardiomyocytes (CMs) need to develop (Figure 1A,B) [8,60,61,62,63,64,65,66]: (1) sarcomere maturation, which includes sarcomere component maturation (an Myh7 to Myh6 and TnI1 to TnI3 switch), and arrangement maturation (disordered and irregular arrangement switched to a rod-like and orderly arrangement); (2) metabolism maturation, which is a shift from anaerobic glycolysis to oxidative phosphorylation due to mitochondrial maturation, with an increase in the number of mitochondrial and their ridges, as well as a closer proximity of mitochondria to sarcomeres; and (3) electrophysiological maturation, in which transverse tubules (T-tubules) form with a gradually increased density and integrity and are accompanied by a significant improvement in calcium handling and excitation–contraction coupling. Failure of CM maturation results in arrhythmias and heart failure.

In terms of sarcomere maturation, several sarcomere components switch from a fetal to an adult isoform: (1) Myosin heavy chain (Myh) switches from fetal MYH7 to adult MYH6 in mice. In contrast, MYH7 is the predominant isoform in the adult heart of humans, and is it already established by 5 weeks of gestation [67,68]. (2) Myosin light chain (MYL) switches from fetal MYL7 (MLC2a) to adult MYL2 (MLC2v) in ventricles. In addition, MYL7 expression becomes restricted to atrial CMs in adults [69,70]. (3) Troponin switches from fetal TNNI1 to adult TNNI3 [71].

In terms of metabolism maturation, in addition to the mitochondria switch, metabolic transcriptional regulators undergo active changes, which include upregulations of genes involved in fatty acid metabolism, oxidative phosphorylation, and mitochondrial biogenesis, such as *Ppargc1a/b*, *Ppara*, *Nrf1/2*, and *Esrra/b/g*, and downregulations of glycolytic genes [72,73]. Glycolytic enzymes switch from fetal HK1 (hexokinase 1) to HK2 (hexokinase 2) [74], and COX (cytochrome c oxidase) subunit 8 switches from fetal COX8A to adult COX8B [75].

In terms of electrophysiological maturation, in addition to T-tubule formation, there are also important switches: (1) The resting membrane potential changes from less negative (approximately −50 to −60 mV) to more negative (approximately −85 mV) because of the higher expressions of the potassium channels (Kir) Kir2.1 and Kir2.2, encoded by genes *KCNJ2* and *KCNJ* [76]. (2) The upstroke velocity action potential switches from slow (approximately 15–30 V/s) to fast (approximately 70–242 V/s) due to the higher expressions of *SCN5A* and other sodium channels [77]. (3) The plateau phase of the action potential switches from short to long due to the higher expression of the Cav1.2 core component CACNA1C and alternative splicing of its auxiliary subunit CACNB2 [78].

However, the mechanisms that govern CM maturation remain elusive. It is now generally believed that microenvironmental factors and intracellular transcriptional regulation, as well as post transcriptional modifications, play important roles. For example, mature primary cultured CMs gradually lose their maturity, and immature CMs transferred to adult hearts can mature [79,80], indicating that the microenvironment surrounding CMs plays an important role in the regulation of maturation. In addition, geometric constraints, extracellular matrix viscoelasticity, mechanical strain, and electrical stimulation are currently used to promote CM maturation in vitro [81,82,83,84], thus providing evidence that microenvironmental factors play a role in cardiomyocyte maturation. However, due to the largely unknown mechanisms by which microenvironmental factors regulate CM maturation, the maturity of CMs obtained through environmental factor stimulation is much lower than that of mature adult cardiomyocytes. Furthermore, biochemical and transcriptional factors have also been suggested to play critical roles in CM maturation [85,86,87,88,89,90,91,92]. We summarize the CM maturation factors that have been reported and those that remain unknown in Figure 1C.

**Figure 1 biology-13-00234-f001:**
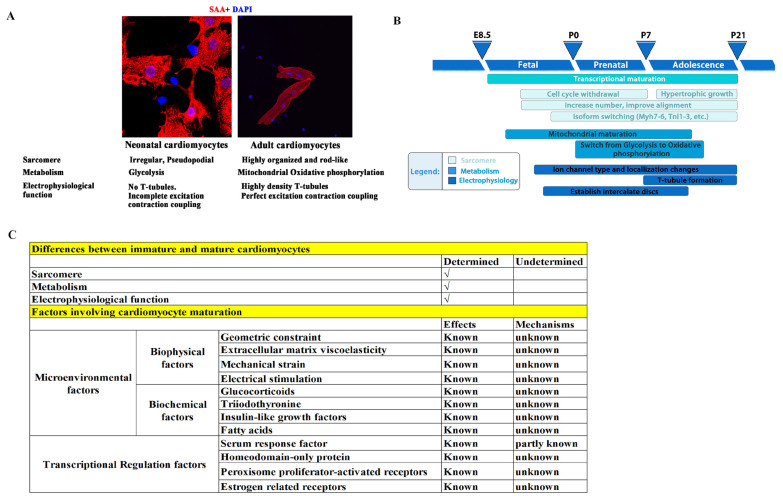
Illustration of the three main characteristics that immature CMs require for development. (**A**) Neonatal and adult CMs have differences in sarcomere, metabolism, and electrophysiological functions. (**B**) Three main characteristics that immature CMs must develop, and the presumed key time points of these maturation events in rodents. Redrawn from [12]. (**C**) Summary of cardiomyocyte maturation factors that have been reported and those that remain unknown [67,68,69,70,71,72,73,74,75,76,77,78,81,82,83,84,85,86,87,88,89,90,91,92].

### 3.2. Immature Lungs in Children

There are five developmental stages of human lungs (Figure 2A) [93,94,95,96]: (1) The embryonic stage (pulmonary bud stage)—occurring between the third and sixth weeks of gestation, this stage marks the appearance of precursor cells and epidermal cells known as pulmonary buds. (2) The pseudoglandular stage (bronchial stage)—taking place between the sixth and sixteenth weeks of gestation, this stage is characterized by bronchial formation. The pulmonary buds proliferate and migrate to the visceral mesenchyme, forming airways and peripheral acinar buds, and eventually developing into alveoli. (3) The tubular phase (alveolar tubular phase)—this stage occurs from 16 to 26 weeks of gestation and involves the differentiation of intra-airway epithelial cells, including basal cells, cupped cells, ciliated cells, and other secretory cells. (4) The vesicular phase (terminal vesicular phase)—spanning from 26 to 36 weeks of gestation, the differentiation of peripheral glandular alveoli characterizes this phase, including cubic type 2 alveolar (AT2) cells and squamous type 1 alveolar (AT1) cells. At this stage, alveoli gradually expand and are covered by AT1 cells, and AT2 cells differentiate to produce alveolar surfactant. (5) The alveolar phase—this phase, which extends from 36 weeks of gestation to adolescence, is the main stage of alveolar formation, with over 90% of alveoli forming at ages 0–7 years. Bronchopulmonary dysplasia, a major complication of prematurity, occurs when the immature lungs fail to develop due to various damaging factors such as hyperoxia, toxicity, and inflammation. Therefore, the lungs of children are immature and in a critical stage of alveolar development, while adult lungs have completed their development. Consequently, many drugs effective in treating adult lung diseases have limited efficacy in addressing lung diseases in infants and children [97,98].

Postnatal lung maturation includes two aspects of maturation: (1) Alveoli maturation, in which the number of alveoli in humans increases from 0 to 50 million at birth to >300 million at adulthood [99,100]. (2) Microvascular maturation, in which the double capillary networks in the parenchymal septa at birth are restructured to a single capillary system at adulthood to enhance gas–blood exchange (Figure 2B) [101,102]. Although the significant differences between the lungs of children and those of adults have been clearly established, the mechanisms underlying postnatal lung maturation are far from clear. Transcriptomic analysis has revealed that postnatal lung maturation can be divided into four different stages (stage 1–4), exhibiting a two-phase pattern of angiogenesis, alveolarization, and neurogenesis. For example, in mice, compared to stage 1 (postnatal day 0–3), stage 2 (postnatal day 4–7) has shown a significant upregulation of angiogenic genes; compared to stage 3 (postnatal day 9–12), there was a peak in the expression of angiogenic genes in stage 4 (postnatal day 13–18) [103]. The transcriptomic analysis of postnatal lung development has suggested that there might be a complicated and extensive genomic coordination between vascular, neurogenic, and alveolar processes for lung maturation [103]. A more recent combined genomic, epigenomic, and biophysical analysis has identified that AT1 cells have a distinct signaling hub that integrates Shh/Wnt/Pdgf signaling pathways for early postnatal alveoli formation [96]. Several important pathways regulating postnatal lung maturation have also been suggested [104,105,106,107,108,109,110,111]. However, many important questions need answers. For example, how is the maturation of blood vessels, alveoli, and nerves coordinated and regulated? Why are some premature infants unable to develop mature lungs, despite the strong regenerative ability of alveoli throughout childhood and even adulthood? The answers to these questions will help us to treat the various forms of lung damage inflicted early in life. We illustrate the known lung maturation factors and certain key pathways regulating postnatal lung maturation in Figure 2C.

**Figure 2 biology-13-00234-f002:**
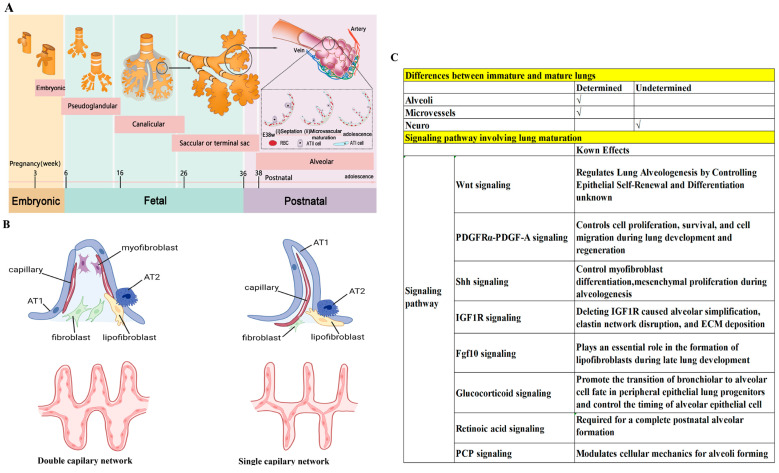
Differences between immature and mature lungs. (**A**) Five developmental stages of human lungs. Note that 90% of alveoli are formed at ages 0–7 years. (**B**) Different capillary network between immature and mature lungs. Redrawn from [101]. (**C**) Several pathways involved in lung maturation [99,100,101,102,103,104,105,106,107,108,109,110,111].

## 4. Creation of Neonatal Surgical Rodent Models of CHDs

### 4.1. Challenges in Constructing Neonatal Surgical Rodent Models of CHDs

The lack of neonatal rodent models for cardiac VO, PO, IPF, and RPF limits our understanding of how hemodynamics regulates postnatal heart and lung development [23,28,30,31]. Consequently, managing CHD primarily relies on surgical intervention, and thus the long-term quality of life for children with complex CHD remains poor [112,113]. The challenges of creating neonatal rodent CHD models are as follows: (1) the size of the neonatal rodent heart is extremely small, with limited operation space; (2) neonatal anesthesia is ice cooling, limiting the surgical time to no more than 20 min; and (3) cannibalization may account for additional post-surgical mortality in neonatal surgery. The above three challenges require superb cardiac microsurgical skills.

As the largest pediatric heart center in the world, the heart center of Shanghai Children’s Medical Center has about 4000 open-heart surgeries per year; as a result, surgeons at Shanghai Children’s Medical Center are very skillful in cardiac surgery. Thus, in recent years, they have successfully developed a series of neonatal CHD models (Figure 3), including models of nACF, nPAB, nPVB, and nTAC [23,28,30,31,114].

### 4.2. Key Points of Constructing Neonatal Surgical Rodent Models of CHDs

#### 4.2.1. nACF Surgery

nACF surgery [33] (Figure 3A,B) includes the following four steps: (1) anesthesia and fixation; (2) abdominal aorta (AA) and inferior vena cava (IVC) exposure; (3) fistula creation, and (4) abdominal closure. There are two key points of nACF surgery: (1) During fistula creation, a mixing of arteriovenous blood in IVC should be observed under a microscope to make sure a successful fistula was created. (2) When exposing the AA and IVC, the small intestine and large intestine should be slowly moved with a cotton swab to prevent bleeding.

#### 4.2.2. nPAB Surgery

nPAB surgery [30] (Figure 3C,D) includes the following four steps: (1) anesthesia and fixation; (2) pulmonary artery (PA) exposure; (3) PA banding; and (4) closure of the thoracic cavity. There are two key points of nPAB surgery: (1) The PA should be exposed with an incision as small as possible to reduce the total time for the thoracotomy surgery, which is critical for the neonatal pups’ recovery from anesthesia. (2) The PA should be separated from its adjacent small vessels with a blunt needle to avoid bleeding.

#### 4.2.3. nTAC Surgery

nTAC surgery [32] (Figure 3E,F) includes the following four steps: (1) anesthesia and fixation; (2) ascending aorta exposure; (3) aorta banding; and (4) closure of the thoracic cavity. There are two key points of nTAC surgery: (1) Blunt needles should be used to prevent intraoperative bleeding, which is a primary cause of postoperative mortality. (2) During the surgery, the pericardium must be carefully separated from the aortic surface to expose the aortic site for banding. The pericardium should be kept intact as much as possible to avoid postoperative adhesion.

#### 4.2.4. nPVB Surgery

The nPVB surgery [28] (Figure 3G,H) includes the following four steps: (1) anesthesia and fixation; (2) pulmonary vein (PV) exposure; (3) PV banding; and (4) closure of the thoracic cavity. There are two key points of nPVB surgery: (1) The right upper and right middle PVs, which are thicker than the other PVs, are selected for banding to replicate PVS. (2) The chest is closed layer by layer to avoid postoperative pneumothorax.

In summary, nACF surgery, which does not require opening the thoracic cavity, is easier to perform than the other three neonatal surgeries. All neonatal surgeries require ice-cooling anesthesia to reduce the operation time as much as possible, which increases the rate of neonatal pup recovery from anesthesia. A blunt needle is required to avoid intraoperative bleeding. For thoracotomy surgery, preventing postoperative adhesions and pneumothorax is key for the neonatal pups’ long term survival. Surgical videos for each neonatal cardiac surgery can be found at public access websites, which are summarized in Table 2.

## 5. Hemodynamic Melody of Pediatric Heart and Lung Development

### 5.1. Cardiomyocyte Proliferation

Promoting cardiomyocyte proliferation is a fundamental method for treating heart failure that not only improves heart failure caused by myocardial infarction [117,118], but also improves pediatric heart failure caused by CHDs [78,89]. A previous study has demonstrated that children with TOF have impaired RV cardiomyocyte proliferation due to a reduced expression of epithelial cell transforming 2 (*ECT2*) [119]. The study also found that beta blockers can promote cardiomyocyte proliferation by enhancing the expression of *ECT2* [119]. Thus, beta blockers have been recommended to treat TOF [119]. The New England Journal of Medicine has highlighted the study, commenting that the treatment of CHDs may not rely on surgical correction; this study opened a new approach for treating CHDs [120].

However, this study lacked a nPAB model to match the most important TOF clinical feature—RVOT obstruction, which produces RV PO–and made its conclusion based on an increase in the percentage of polyploidic cardiomyocytes in children with TOF [119]. An increased percentage of polyploidic cardiomyocytes could be a consequence of cardiomyocyte proliferation (Figure 4). Increasingly, more studies have shown that PO promoted neonatal cardiomyocyte proliferation in both RV and LV [31,121,122,123,124,125]. We have also found that PO promoted RV cardiomyocyte proliferation in children with TOF [31]. Therefore, it is inappropriate and unwise to ignore the contribution of PO to cardiomyocyte proliferation when studying pediatric TOF. Moreover, CHD is a polygenic disease, and mutations of *ECT2* have rarely been reported in TOF patients [119]. Thus, it is more reasonable to open a new approach for CHD treatment focused on abnormal hemodynamics than a single gene modification.

Nevertheless, the effect of PO on promoting proliferation of cardiomyocytes decreases with age, and PO cannot re-induce proliferation in differentiated and mature CMs at adult [31]. However, interestingly, VO re-induces proliferation in prepubertal cardiomyocytes in both the LV and RV [22,23], but does not promote proliferation in neonatal cardiomyocytes [126]. In contrast, with heart failure and at an adult stage, pressure unloading promotes cardiomyocyte proliferation [127]. These results show the complex roles of forces generated by abnormal hemodynamics on cardiomyocyte proliferation.

Force is one of the foundations of matter [128,129], and the maintenance of many life phenomena depend on a balance of forces [128,129]. How force regulates cardiac regeneration remains elusive, and perhaps changing a single mechanoreceptor can promote cardiac regeneration. In fact, a recent study has demonstrated that Plxnd1 is a necessary and sufficient condition for endothelial cells to sense force for regulating cardiovascular pathophysiology [130]. We have also found that Plxnd1 significantly increased in prepubertal cardiomyocytes under VO conditions (Figure 5). The introduction of neonatal ventricular PO and VO models provides a platform for researchers to explore the regulation of cardiac regeneration by forces, and the role of Plxnd1 may be one of the future directions. In addition to Plxnd1, other mechanoreceptors, such as YAP1, Sdc4, and Itgal1, have also been implicated in the PO and VO regulation of postnatal cardiac development (Figure 5). It should be noted that combinations of mechanical receptors undergo changes at different developmental stages and under different forces (VO or PO).

### 5.2. Cardiomyocyte Maturation

Maturation is the most important event of postnatal CM development in mammals [62,131]. Although the underlying mechanisms of cardiomyocyte maturation are largely unknown, the switch from pediatric to adult hemodynamics is undoubtedly an important contribution [132,133]. Due to the increase in body size, the preload and afterload faced by the adult heart are significantly greater than that of children [132,134]. In other words, it may be the force generated by the increasing preload and afterload that guide the maturation of cardiomyocytes.

The immaturity of CMs derives from induced pluripotent stem cells (iPSC-CMs) and their failure to mature in vivo limits their clinical use [131,132]. Therefore, in vitro experiments with various stimuli, including electrical rhythm, force, and 3D culture, were used to promote the maturation of iPSC-CMs [131,132]. Although their maturity has increased, there is still a large gap between stimuli-enhanced iPSC-CMs and mature adult CMs [131,132]. The differences between in vivo and in vitro conditions, as well as an insufficient understanding of force, may account for the limited effect of force on CM maturation in studies.

Recent studies have indicated that CM proliferation and maturation are two opposite processes [134,135]. Because VO and PO promote CM proliferation at neonatal or prepubertal stages [23,31], it is not surprising to find that CM maturation was impaired because of VO [22,23]. We have also found that VO impeded CM maturation (Figure 5). The purpose of iPS cm transplantation is to treat patients with heart failure, the hemodynamic characteristic of which is VO. Thus, the finding that VO impedes CM maturation may further deter the transplantation of iPSC-CMs into patients with heart failure.

Another issue is whether VO or PO affect CM maturation temporarily or permanently. In other words, when VO or PO are released, can the CMs still mature? This is the situation that exists in children with CHD. When the structural defects of the hearts of children with CHDs are corrected, will the maturity of their adult CMs be affected? Clinical investigations have found that even with perfect anatomic correction in childhood, patients with TOF are still at a high risk of arrhythmia [136], a feature of cardiomyocyte immaturity, suggesting that CM maturation may be permanently impaired.

In summary, some force is helpful for CM maturation in vitro, and the force generated by age-increased preload and afterload may guide CM maturation in vivo. However, excessive force due to VO and PO impairs postnatal cardiomyocyte maturation. A key question is how force regulates CM maturation. Does maturation share the same mechanoreceptor(s) used in proliferation? After mechanoreceptor activation, different pathways involved in maturation and proliferation may be subsequently activated. In fact, we have found that VO initiated an immune response at neonatal and prepubertal stages in both the RV and LV, including macrophage activation [22,23,126]. Consistently, the activation of the mechanoreceptor Plxnd1 in endothelial cells has led to expression of macrophage chemokines, which recruited macrophages from the peripheral blood to the heart [130]. These interesting studies are a good foundation for us to understand how force, a basic component of the material world, affects cardiovascular pathophysiology. We summarize the genes and pathways identified from our omics studies in Figure 5, which shows that considerable further investigation and verification are still needed.

### 5.3. Lung Development

The interplay of the heart and lung profoundly, functionally, and anatomically determine a person’s quality of life [137,138,139]. About half a century ago, an autopsy study revealed a decrease in lung volume and pulmonary dysfunction in patients with RPF-associated CHDs [140]. However, the underlying mechanisms remained elusive until recent nPAB surgery was developed, which showed that RPF caused alveolar dysplasia, angiogenesis impairment, and inflammation [28,29]. RPF also impaired cell–cell communication and axon guidance, two critical events of late alveolar formation [29,103]. Axon guidance is required for the coronavirus infection, which may explain why children with RPF-associated CHDs are relatively insensitive to COVID-19 infection [29,141]. RPF reduces the intravascular pressure and gas exchange rate, which means a reduced force in the vessels of the lung. It is possible that force is the basic cause of RPF-induced pulmonary dysplasia. Interestingly, RPF also induces inflammation [25], similar to that of VO-induced cardiomyocyte proliferation, further suggesting that there may be a force-mediated regulation.

Current bronchopulmonary dysplasia (BPD) animal models for premature infants include models of hyperoxia, pulmonary ventilation, and lipopolysaccharide [96,142], all of which aim to induce inflammation, yet yield poor targets for improving lung development of premature infants [94,143]. This may be because inflammation is not the initiating factor. Complications of premature infants often include PH and PDA [94,144], both of which induce RPF. Thus, RPF may account for premature BPD, and the nPAB model may provide a new window into the study of BPD.

In contrast to RPF, IPF, which increases intravascular pressure, leads to pulmonary congestion and thickening of the pulmonary small blood vessels, a characteristic of PH, which in turn increases intravascular pressure, and ultimately a vicious circle forms. A nPVB model has shown a similar presentation as children with PVS, which included PV thickening, pulmonary small vessels thickening, pulmonary congestion, PH, and RV failure [28]. Consistently, the nACF model causing IPF also showed the thickening of pulmonary small blood vessels, but to a lesser extent [33]. Mechanoreceptors are less studied in the lungs than in the heart, including their regulation of force in lungs.

## 6. Summary and Prospects

Force is a basic component of our world and is generated by abnormal hemodynamics in CHDs, producing profound effects on postnatal heart and lung development, which we were previously unaware of due to the lack of neonatal rodent animal models. Nevertheless, we now know that both VO and PO promote prepubertal CM proliferation and impede CM maturation, and we also know that RPF leads to pulmonary dysplasia and IPF leads to the thickening of pulmonary blood vessels.

### 6.1. Clinical Decision Making

VO- and PO-impaired CM maturation are associated with arrhythmias and weakened cardiac systolic function [132,133], suggesting that childhood corrections of VO or PO or the promotion of CM maturation may improve adult cardiac performance. However, VO and PO promote prepubertal CM proliferation, which is fundamental to heart failure treatment [31,33], suggesting that VO and PO should be enhanced. Clearly, these two suggestions are contradictory. An in-depth understanding of how force regulates CM proliferation and maturation is a prerequisite for us to precisely regulate proliferation and maturation via VO and PO to improve the quality of life of children with CHD.

RPF and IPF are both detrimental to pulmonary performance, suggesting that they should be corrected as early as possible. If RPF or IPF cannot be corrected, mechanoreceptor blockers, axon guidance molecules, or vessel thickening inhibitors are suggested to be used as early as possible to improve pulmonary performance of children with CHD.

### 6.2. Limitations and Future Directions

Currently, the neonatal surgical rodent models only help us obtain a very primitive observation, leaving an abundance of unanswered questions: (1) Do mechanoreceptors for VO or PO promote cardiomyocyte proliferation or impede CM maturation? Apart from Plxnd1, other mechanoreceptors have also been revealed. Sdc4 is upregulated, while Itga11 is downregulated in the neonatal PO RV (Figure 5). The expression patterns of neonatal Sdc4 and Itga11 are different from that of adults, in whom both are upregulated [145,146]. How PO or VO regulate CM proliferation and maturation via Plxnd1, Sdc4, and Itga11 might be a future direction of research. Whether there are other mechanoreceptors that play a crucial role in regulating postnatal heart and lung development also needs to be explored. (2) Metabolic reprogramming has also occurred under the condition of neonatal PO and VO (Figure 5). Is metabolic reprogramming the cause or result of CM proliferation? How do PO and VO initiate metabolic reprogramming of CMs? (3) The chromatin openness of many genes associated with heart and lung development has been changed greatly by PO, VO, IPF, and RPF. The epigenetics study of hemodynamics should also be a future direction.

## 7. Conclusions

Abnormal hemodynamics and cardiac and pulmonary development intertwine to form the most important features of childhood CHDs. With the development of neonatal CHD rodent models, a window into the CHD-associated hemodynamic regulation of postnatal heart and lung development has been opened, allowing us a superficial but important view of postnatal heart and lung development (Figure 5). With continuous and in-depth research on neonatal CHD rodent models, the long-term quality of life of children with CHD may be improved, and the mechanisms underlying CM proliferation and maturation and lung development will be more comprehensively understood.

## 8. Searching Methods

We used “congenital heart disease AND neonatal OR animal models OR cardiomyocyte OR proliferation OR maturation OR alveoli” as the search method in PubMed.

## Figures and Tables

**Figure 3 biology-13-00234-f003:**
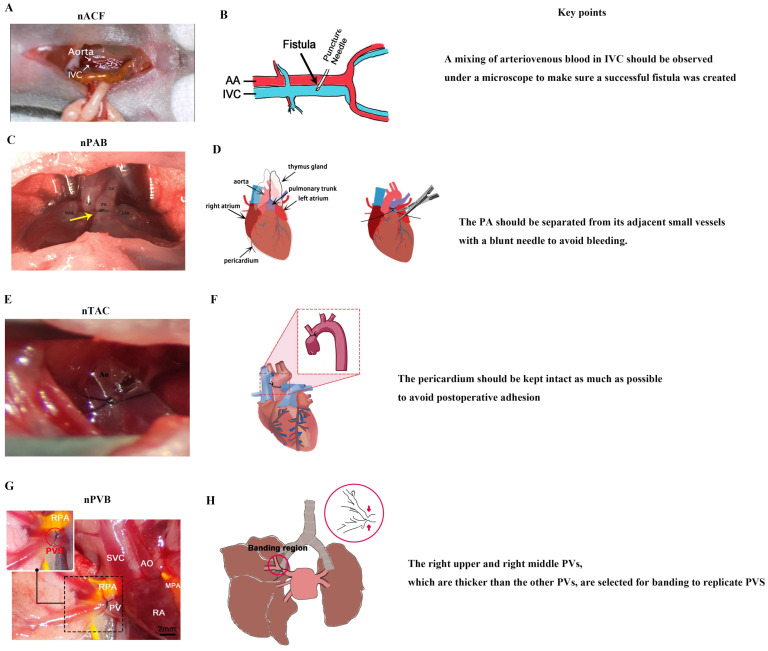
Neonatal surgical rodent models of CHDs. (**A**) nACF produces RA VO, RV VO, LA VO, LV VO, and IPF. (**B**) Surgical diagram of nACF. (**C**) nPAB produces RV PO and RPF, yellow arrow indicates the banding region. (**D**) Surgical diagram of nPAB. (**E**) nTAC produces LVPO. (**F**) Surgical diagram of nTAC. (**G**) nPVB produces PVS and IPF. (**H**) Surgical diagram of nPVB, red arrow indicates the banding region. IVC: inferior vena cava; PA: pulmonary artery; DA: ductus arteriosus; Ao: aorta; RPA: right pulmonary artery; PV: pulmonary vein; SVC: superior vena cava; RA: right atrium; MPA: main pulmonary artery. (**A**) was adopted ref. [23] under a Creative Commons Attribution-NonCommercial-NoDerivs License; (**B**) was adopted from ref. [115] under a Creative Commons Attribution-NonCommercial-NoDerivs License; (**C**) was adopted from ref. [30] with the permission of the publisher; (**D**) was adopted from ref. [116] with the permission of the publisher; (**H**) was adopted from ref. [28] under a Creative Commons Attribution-NonCommercial-NoDerivs License).

**Figure 4 biology-13-00234-f004:**
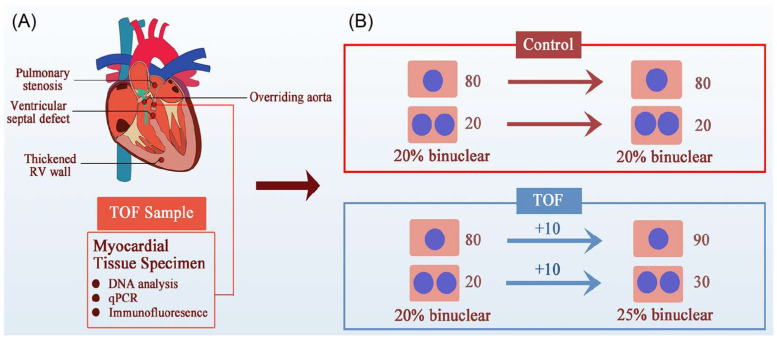
An alternative possibility for the increased percentage of polyploid cardiomyocytes in TOF children. (**A**) The four clinical features of TOF are RVOT obstruction, ventricular septal defect, thickened RV, and aortic riding, the most important of which is RVOT obstruction, which leads to RV PO and RV failure. (**B**) Illustration of the increased binuclear cardiomyocytes generated in TOF patients. Greater numbers of binuclear cardiomyocytes do not indicate a failure of cytokinesis. For example, in the beginning of a study, if both the control group and TOF group had 80 mononucleated and 20 binucleated cardiomyocytes (CMs), the proportion of binucleated CMs was 20%. Under PO conditions, both mononucleated and binucleated CMs in the TOF group increased by 10, and the proportion of binucleated CMs was 25%. Therefore, an increase in the proportion of binucleated CMs does not necessarily mean impaired cytokinesis and proliferation.

**Figure 5 biology-13-00234-f005:**
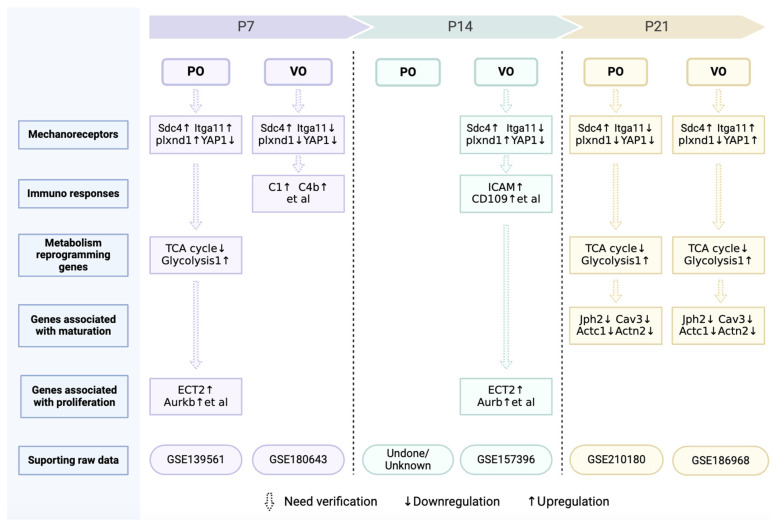
Omics information on postnatal cardiac development under of VO or PO.

**Table 1 biology-13-00234-t001:** Abnormal hemodynamics classifications and corresponding congenital heart diseases (CHDs).

Abnormal Hemodynamics Classification	RV VO	LV VO	RV PO	LV PO
CHD	ASD, PDA, PAPVD, VSD, TR, EA, PVI, TOF, AVSD, PAH	ASD+TS/RVOT, MR, MVI, AR, BAV	PVS, VSD, PDA, PAH, MS, AS, TOF, IAA, CAA, PAS, PVS	BAV, IAA, CAA, MS, AS, HCM

**Table 2 biology-13-00234-t002:** Neonatal cardiac surgery teaching videos.

Surgery	Website	Ref.
nACF	https://www.ahajournals.org/doi/suppl/10.1161/JAHA.121.020854, accessed on 13 August 2021	[23]
nTAC	https://www.nature.com/articles/s41596-020-00434-9, accessed on 16 December 2020	[32]
nPAB	https://www.jtcvs.org/article/S0022-5223(17)31192-3/fulltext#supplementaryMaterial, accessed on 14 June 2017	[30]
nPVB	https://cellandbioscience.biomedcentral.com/articles/10.1186/s13578-023-01058-8, accessed on 19 June 2023	[28]

## Data Availability

The raw data in Figure 5 were all generated by us and were deposited in NCBI’s Gene Expression Omnibus database (https://www.ncbi.nlm.nih.gov/geo, accessed on 19 August 2020, 8 December 2021, 16 August 2021, 1 May 2024, and 7 April 2022) with accession number GSE139561, GSE180643, GSE157396, GSE210180, and GSE186968.

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
