# Peer review of "Hemodynamic Melody of Postnatal Cardiac and Pulmonary Development in Children with Congenital Heart Diseases"

_biology, 2024, doi:10.3390/biology13040234_

Round 1

Reviewer 1 Report

Comments and Suggestions for Authors

The authors have provided a comprehensive summary of the pathophysiology of heart and lung development in patients with Congenital Heart Defects (CHD). I found the review manuscript to be very informative. However, the current version seems more like a textbook chapter than a review article. Here are some recommendations for improvement:

1) The Simple Summary seems to duplicate the Abstract. This should be revised.

2) There is a contradiction in the summary regarding neonatal rodent models of CHDs (page 1, line 12 and line 15).

3) Each section should discuss what has been reported and what remains unknown. It would be helpful to include a schematic figure in each section to summarize the key points. It is recommended to replace Figures 5, 6, 7 and 8 with the schematic figures to summary the genes and pathways identified from -omics studies.

4) Figure 4 does not explain the cellular or molecular mechanisms of polyploidy in CHD patients.

5) There appears to be a typo in Section 5 ("Hemodyn").

6) It is unclear how the authors selected publications to be discussed in this review. Please include the information on the keywords that were used to search for literature in PubMed (or indicate the database if not PubMed) to be included in this review article.

Overall, these changes would enhance the clarity and organization of the article.

Comments on the Quality of English Language

No issue.

Author Response

Q: The authors have provided a comprehensive summary of the pathophysiology of heart and lung development in patients with Congenital Heart Defects (CHD). I found the review manuscript to be very informative. However, the current version seems more like a textbook chapter than a review article. Here are some recommendations for improvement:

A: Thank you for the nice comments.

Q:1) The Simple Summary seems to duplicate the Abstract. This should be revised.

A:We revised the simple summary as follows:

Abnormal hemodynamics and cardiac and pulmonary development intertwine to form the most important features of children with congenital heart diseases (CHDs). Here, we review the varieties of hemodynamic abnormalities that exist in children with CHDs and the recently developed neonatal rodent models of CHDs, which have inspired new ideas in cardiomyocyte proliferation and maturation, and in alveolar development.

Q:2) There is a contradiction in the summary regarding neonatal rodent models of CHDs (page 1, line 12 and line 15).

A:Thank you for the comments. We deleted the sentence at line 12 “However, due to the lack of neonatal rodent models of CHDs, it is unclear how CHD-associated hemodynamics shape postnatal cardiac and pulmonary development.”

Q:3) Each section should discuss what has been reported and what remains unknown. It would be helpful to include a schematic figure in each section to summarize the key points. It is recommended to replace Figures 5, 6, 7 and 8 with the schematic figures to summary the genes and pathways identified from omics studies.

A: Thank you for the helpful suggestions.

We discussed the information that has been reported and the information that remains unknown in each section, and we added a schematic figure in each section to summarize the key points.

Accordingly, schematic figures summarizing the genes and pathways identified from omics studies were added to replace Figures 5, 6, 7, and 8.

Q:4) Figure 4 does not explain the cellular or molecular mechanisms of polyploidy in CHD patients

A:Thank you for the comments.

We replaced the caption of Figure 4 with An alternative possibility for the increased percentage of polyploid cardiomyocytes in TOF children.

Q:5) There appears to be a typo in Section 5 ("Hemodyn")

A:We corrected it to “hemodynamics”.

Q:6) It is unclear how the authors selected publications to be discussed in this review. Please include the information on the keywords that were used to search for literature in PubMed (or indicate the database if not PubMed) to be included in this review article.

A:Thank you for the suggestion.

We used congenital heart disease AND neonatal OR animal models OR cardiomyocyte OR proliferation OR maturation OR alveoli as the search method in PubMed. We added this information at the end of the manuscript.

Q:7) Overall, these changes would enhance the clarity and organization of the article.

A:Thank you again for these helpful comments.

Reviewer 2 Report

Comments and Suggestions for Authors

Comments to the Author:

In this study, the authors reviewed the varieties of hemodynamic abnormalities that exist in children with CHDs, the recently developed neonatal rodent models of CHDs, and the inspirations these models in cardiomyocyte proliferation and maturation, and in alveolar development. The authors also stated the current limitations, future directions, and clinical decision-making based on their summary of previous studies.

It is a straightforward manuscript and an important addition to the literature. My comments on this study are below.

Major comments:

1.    For line 146-157, in terms of the sarcomere maturation in cardiomyocytes, it will be more informative to provide some information about the sarcomeric actin components. What actin isoforms are the dominant type in the sarcomere at different developmental stages. Is there an actin isoform switch during the cardiomyocyte maturation? What proteins are involved in these changes?

2.    Fig.1B shows interesting information but was never mentioned in the text. In addition, is this scheme drew based on the studies in mouse? Authors should also cite the original studies in the figure legend.

3.    In line 301-308, mechanical force is one of the major factors to drive tissue development, regeneration and homeostasis. Numerous signaling pathways have been well studied of able to sense extracellular force and transduce it to biological activities in cells. Hippo signaling pathway is one of them. In addition to Plxnd1, could author state more candidate genes/pathways? It seems unlikely that a single molecule is capable to regulate cardiac regeneration which is a quite complicated process involving many cellular and molecular events to coordinate.

4.    It’s unclear when the figure legend said that the raw data have been deposited in NCBI. Is the data in Fig.6 generated in this study? Or the authors re-analyzed the data in an online database? Same concerns for Fig. 7.

Minor comments:

The font size in Fig.6 is inconsistent.

Comments on the Quality of English Language

Minor editing of English language is required to reach the publication standards.

Author Response

Q:In this study, the authors reviewed the varieties of hemodynamic abnormalities that exist in children with CHDs, the recently developed neonatal rodent models of CHDs, and the inspirations these models in cardiomyocyte proliferation and maturation, and in alveolar development. The authors also stated the current limitations, future directions, and clinical decision-making based on their summary of previous studies.

It is a straightforward manuscript and an important addition to the literature. My comments on this study are below.

A:Thank you for the nice comments.

Q:1.For line 146-157, in terms of the sarcomerematuration in cardiomyocytes, it will be more informativeto provide some information about the sarcomeric actin components. What actin isoforms are the dominant type in the sarcomere at different developmental stages. Is there an actin isoform switch during the cardiomyocyte maturation? What proteins are involved in these changes?

A:Thanks for the suggestion. We added the information regarding the sarcomere actin component switch during the maturation transition.

Q:2.Fig.1B shows interesting information but was nevermentioned in the text. In addition, is this scheme drewbased on the studies in mouse? Authors should also cite the original studies in the figure legend.

A:Thanks for the comments. We added the information in the text and their associated references.

Q:3.In line 301-308, mechanical force is one of the majorfactors to drive tissue development, regeneration andhomeostasis. Numerous signaling pathways have been well studied of able to sense extracellular force and transduce it to biological activities in cells. Hippo signaling pathway is one of them. In addition to Plxnd1,could author state more candidate genes/pathways? It seems unlikely that a single molecule is capable to regulate cardiac regeneration which is a quite complicated process involving many cellular and molecular events to coordinate.

A:Thank you for the comments.

We added this information in the text, and discussed the Hippo signaling pathway and other pathways in the new version.

Q:4.It’s unclear when the figure legend said that the rawdata have been deposited in NCBI. Is the data in Fig.6generated in this study? Or the authors re-analyzed the data in an online database? Same concerns for Fig. 7.

A:Thank you for the comments.

We reanalyzed the data in an online database, and the raw data are what we deposited in the NCBI database. We added this statement in the Data Availability Statement.

Round 2

Reviewer 1 Report

Comments and Suggestions for Authors

Thank you for addressing my comments. Two more issues to be addressed:

1) page 1 line 14, please correct the grammar "which have inspiredbrought us new ideasinspirations in cardiomyocyte proliferation and maturation".

2) page 6 line 219, please delete "of maturation".